# Summer Distributional Characteristics of Surface Phytoplankton Related with Multiple Environmental Variables in the Korean Coastal Waters

Ji Nam Yoon [1,2], Minji Lee [1], Hyunkeun Jin [3], Young Kyun Lim [1,2], Hyejoo Ro [1], Young Gyu Park [3] and Seung Ho Baek [1,2,*]

1   Risk Assessment Research Center, Korea Institute of Ocean Science and Technology, Geoje 53201, Korea; dbswlska@kiost.ac.kr (J.N.Y.); mjlee@kiost.ac.kr (M.L.); limyk0913@kiost.ac.kr (Y.K.L.); hyejooro97@gmail.com (H.R.)
2   Department of Ocean Science, University of Science and Technology, Daejeon 34113, Korea
3   Ocean Circulation Research Center, Korea Institute of Ocean Science and Technology, Busan 49111, Korea; hkjin@kiost.ac.kr (H.J.); ypark@kiost.ac.kr (Y.G.P.)
*   Correspondence: baeksh@kiost.ac.kr

**Abstract:** Multiple environmental variables related to ocean currents, freshwater runoff, and upwelling in a coastal area have complex effects on the phytoplankton community. To assess the influence of environmental variables on the phytoplankton community structure during the summer of 2019, we investigated the various abiotic and biotic factors in Korean coastal waters (KCWs), separated into five different zones. Summer environmental factors in KCWs were strongly influenced by Changjiang Diluted Water (CDW) in St. SO (Southern Offshore) 1 and 2, upwelling in St. SI (Southern Inshore) 2–4, and Nakdong River discharge in St. SI 12. In particular, low–salinity water masses ($p < 0.05$ for nearby locations) of CDW gradually expanded from the East China Sea to southwestern KCWs from June to July. In addition, there were high levels of nutrients following freshwater runoff from the Nakdong River in southeastern KCW, which led to the dominance of *Cryptomonas* spp. (81%), a freshwater and brackish water algae. On the other hand, upwelling areas in southwestern KCW were dominated by diatoms *Skeletonema* spp., and are characterized by high phosphate concentrations ($p < 0.05$) and low temperatures ($p < 0.05$) compared to nearby locations. *Leptocylindrus danicus* (20%) was dominant due to the effect of water temperature in the SE (Southeastern area) zone. Low nutrient concentrations were maintained in the East Sea (dissolved inorganic nitrogen (DIN) = 0.39 ± 0.40 μM; dissolved inorganic phosphate (DIP) = 0.09 ± 0.03 μM) and the Yellow Sea (DIN = 0.40 ± 0.07 μM; DIP = 0.04 ± 0.02 μM), which were characterized by low levels of chlorophyll *a* and dominated by unidentified small flagellates (35, 40%). Therefore, our results indicated that hydro–oceanographic events such as upwelling and freshwater run–off, but not ocean currents, provide nutrients to the euphotic layers of the coastal environment and play important roles in determining the phytoplankton community structure during summer in the KCWs.

**Keywords:** Korean coastal waters; Changjiang Diluted Water (CDW); upwelling; river discharge

## 1. Introduction

Phytoplankton are regarded as a key component of freshwater, estuarine, coastal, and offshore ecosystems because of their role at the base of the food web. The abundance and community structure of phytoplankton are strongly influenced by environmental changes. Among these changes, nutrient loading from river discharge is known to be a crucial factor in increasing the abundance of phytoplankton [1]. In particular, there has been a significant increase in the occurrence of harmful algal blooms (HAB) in coastal and oceanic waters that have adversely affected biodiversity and the marine ecosystem [2]. However, due to a lack of evidence, it is difficult to predict the response of phytoplankton communities to

environmental changes from river discharge and ocean currents. Continued monitoring of the relationship between phytoplankton and environmental factors is important to provide a better understanding of how upwelling, river discharge, and ocean currents can impact marine ecosystems in the face of climate change.

Korean coastal waters (KCWs) are affected by ocean currents including the Jeju Warm Current and the Tsushima Warm Current, branches of the Kuroshio Current with high salinity [1,3,4]. The Yellow Sea (YS) western coastal waters of the Korean peninsula is a shallow semi–enclosed marginal sea (20–90 m) in the western Pacific Ocean, surrounded by Korea and China. This region is known to be very productive and is an important marine fishery production area. There is a strong water stratification in the central YS due to a combination of weak winds and strong solar irradiance during the summer period [5]. In addition, the Changjiang Diluted Water (CDW), formed by Changjiang River discharge (CRD) during the summer monsoon, enters KCWs from the East China Sea. CDW has low salinity and a high temperature, and can affect the physicochemical conditions in the receiving region [6]. The southern Korean coast consists of three major river estuaries. The Yeongsan River drains into the southwest coast, Seomjin drains into the south–central coast, and Nakdong drains into the southeast region of the Korean Peninsula. As a result, large amounts of water discharged from these rivers during the rainy seasons bring seeds of phytoplankton populations, or the nutrients cause markedly large phytoplankton blooms [1]. Therefore, numerous abalone, bivalve (oysters and ark shells), and fish farms have been established in the southern coastal areas of Korea. The East Sea is a typical mid–latitude, marginal, semi–closed sea located in the northwestern Pacific Ocean between the Eurasian continent and Japan. The average water depth of the East Sea is 1700 m, but it has deep basins exceeding 2500 m in depth. The East Sea has a well–defined sub–polar front at approximately 37–40° N [7]. This front is created between a warm water mass from the East Korea Warm Current (EKWC), which branches from the Tsushima Warm Current (TWC), and a cold water mass from the North Korea Cold Current (NKCC), a branch of the Liman Current.

Seasonal monsoons strongly affect the climate of the Korean Peninsula. Summer monsoons significantly affect KCWs with relatively weak south to southeasterly winds and heavy precipitation. In KCWs, nutrient loading from the three major rivers usually peaks in summer, likely due to a large amount of water discharged during the rainy seasons, but little is known about the response of the phytoplankton community composition to inorganic nutrient loading. To investigate the relationship between phytoplankton communities and the characteristics of various water masses in KWCs regarding the appearance of harmful species, we examined the oceanographic features of the KCWs during the summer of 2019. We also assessed the distributional properties of phytoplankton groups, including HABs species related to environmental factors caused by different water masses.

## 2. Materials and Methods

### 2.1. Field Sampling

The established 53 stations were designated into five zones (Zone YS, SO, SI, SE, and ES) according to geographical and hydrodynamic characteristics (Figure 1). Zone YS (Yellow Sea, St. YS 1–6) is located between mainland China and the Korean Peninsula, representing a typical shallow epicontinental sea. Zone SO (Southern Offshore, St. SO 1–11) is located near Jeju Island, and is influenced by CDW. Zone SI (Southern Inshore, St. SI 1–13) is influenced by river discharge, and upwelling occurs frequently. Zone SE (Southeastern area, St. SE 1–6) is influenced by the Nakdong River, which provides freshwater input and nutrient pulses. Zone ES (East Sea, St. ES 1–17) is an offshore area of the East Sea, near Dokdo and Ulleungdo. Field sampling was conducted during the summer of 2019 while onboard the R/V Eardo from 2 June to 5 June (SE and ES) and R/V Onnuri from 2 July to 5 July (SI, SO, and YS). Water temperature and salinity were measured at the surface using a YSI 6600 data sonde (Los Angeles, CA, USA) for the horizontal profiles. Surface water was collected using a bucket at all sampling stations during both sampling times.



For nutrient and chlorophyll *a* (Chl *a*) measurements, water samples were filtered through a glass microfiber filter (GF/F; Whatman, Middlesex, UK), then placed in acid–cleaned polyethylene bottles. The filters for Chl *a*, and water samples fixed with HgCl2 for nutrient analysis were stored at –20 °C. For phytoplankton composition analysis, 0.5 L of water was stored immediately in polyethylene bottles and fixed with Lugol's solution (final concentration: 0.5%).

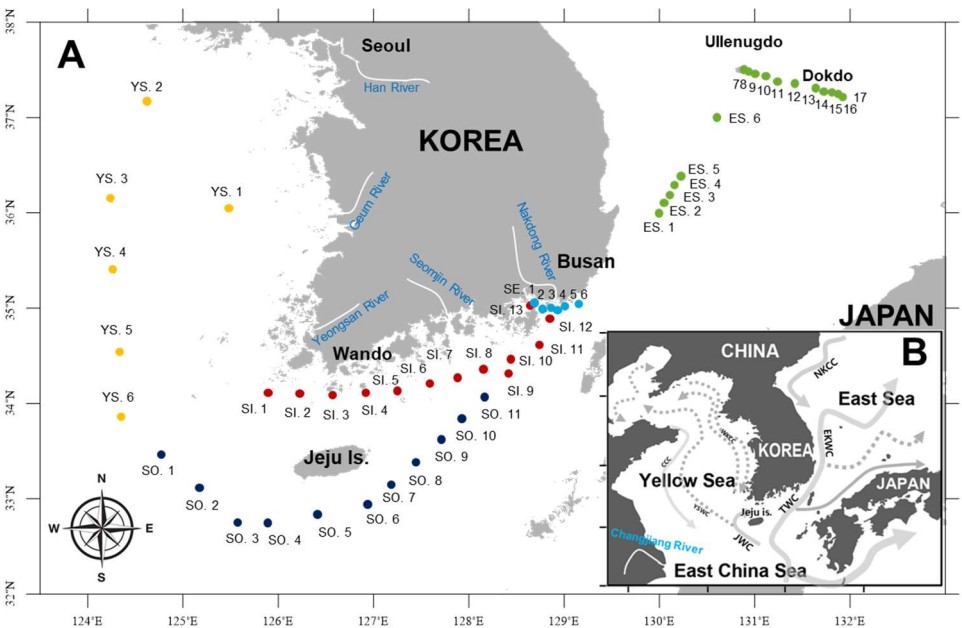

**Figure 1.** Locations of the sampling stations in the Korean Coastal Waters (KCWs) (**A**), with the main water currents (Jeju Warm Current (JWC), Tsushima Warm Current (TWC), East Korea Warm Current (EKWC), West Korea Coastal Current (WKCC), China Coastal Current (CCC), Yellow Sea Warm Current (YSWC) and the North Korea Cold Current (NKCC)) (**B**).

## 2.2. Laboratory Analyses

At the laboratory, Chl *a* was measured using a Turner Designs fluorometer (Turner BioSystems, Sunnyvale, CA, USA) following extraction of filtered material with 90% acetone at room temperature for 24 h in darkness. The concentrations of inorganic nutrients (ammonium, nitrate + nitrite, phosphate, and silicate) were measured using a flow injection auto–analyzer (QuikChem 8000; Lachat Instruments, Loveland, CO, USA). Nutrient concentrations were calibrated using standard brine solutions (CSK Standard Solutions; Wako pure Chemical industries, Osaka, Japan). To count and identify phytoplankton, each 0.5 L Lugol's–fixed sample was concentrated to approximately 50 mL by decanting the supernatant [8]. The concentrated subsamples were gently mixed and then loaded onto a Sedgewick–Rafter counting chamber. Phytoplankton were counted using a light microscope (Carl Zeiss; 37081 Gottingen, Germany) at 200× magnification, and were identified at 400× magnification.

## 2.3. Remote Sensing Analyses

Ocean analysis data of water temperature (SST), salinity (SSS), and sea surface currents for June and July 2019 were provided by the Korea Institute of Ocean Science and Technology (KIOST) [9]. Since March 2017, KIOST has been operating the "Ocean Predictability Experiment for Marine environment (OPEM)", a real–time ocean circulation prediction system in order to detect oceanographic phenomena such as cold water zones and high temperatures occurring in the waters surrounding the Korean Peninsula. OPEM is a regional ocean circulation model developed based on the Modular Ocean Model Version 5, published by the Geophysical Fluid Dynamics Laboratory of the National Oceanic and Atmospheric Administration [10]. The model domain covers the Northwest Pacific

region (99–170° E, 5–63° N). The spatial resolution of the model is 1/24°, making it one of the most high–resolution data sets among ocean numerical models covering the Korean Peninsula and the East China Sea. OPEM applied 12–months of averaged runoff data from the Global River Discharge Database Version 1.1 to simulate freshwater flow into the ocean from rivers such as Changjiang and Yellow in China, and Han, Nakdong, Yeongsan, Geum, and Seomjin in Korea [11]. OPEM's ocean data assimilation system was developed based on the Ensemble Optimal Interpolation method, and through this, ocean observation data were collected in real–time to generate 3D analysis data [12]. SST data assimilation was performed once a day, with the input data from GHRSST Level 4 OSTIA Global Foundation Sea Surface Temperature Analysis, which is SST satellite observation data with a spatial resolution of 0.054° [13]. Data assimilation on the vertical structure of water temperature and salinity was performed once every 7 days with input data from Argo (https://data-argo.ifremer.fr/latest_data/, accessed on 5 June 2019) and in–situ data for temperature and salinity profiles provided by Global Temperature and Salinity Profile Programme (https://www.ncei.noaa.gov/data/oceans/gtspp/realtime/meds_ascii/, accessed on 5 June 2019). A previous study [9] presented the main information of the OPEM system and verification results such as water temperature and salinity in the waters surrounding the Korean Peninsula. Satellite–based Chl *a* observation data used daily Himawari–8 Level 3, and a geostationary orbit satellite operated by the Japanese Meteorological Agency, and the spatial resolution of the data is 1/20° [14].

### 2.4. Meteorological Data

Precipitation and discharge data of the Nakdong, Seomjjin and Yeongsan Rivers were obtained from the Korea Meteorological Administration (KMA, www.weather.go.kr, accessed on 5 June 2019) and National Water Resources Management Information System (WAMIS, www.wamis.go.kr, accessed on 5 June 2019).

### 2.5. Statistical Analysis

A principal component analysis (PCA) was conducted with abiotic and biotic factors to identify significant environmental factors affecting phytoplankton dynamics. The differences in abiotic and biotic factors (including composition of phytoplankton) between the five geographical zones were assessed with a one–way analysis of variance (ANOVA) with Tukey's HSD post-hoc test, and a *t*-test was used to evaluate and confirm differences in environmental factors between stations within a zone. A difference was considered significant when the *p*–value was less than 0.05. All statistical analyses were performed using SPSS version 17.0 (SPSS Inc., Chicago, IL, USA). Canonical correspondence analysis (CCA) was used to estimate the relationships between phytoplankton and environmental variables and visualize the disposition of sampling zones and taxa along the environmental gradients [15]. Therefore, the impacts of measured environmental factors on the occurrences of dominant phytoplankton were determined by CCA using CANOCO version 4.5 for Windows, and phytoplankton abundance and environmental factors were log-transformed to reduce deviations.

## 3. Results

### 3.1. Precipitation and Discharge of the Three River to KCWs

Monthly discharge and precipitation had generally similar seasonal patterns, with low values during the dry season and high values during the rainy season (Figure 2). During the rainy season, the precipitation for June and July in the Nakdong River was 324.3 and 358.9 mm, Seomjin River was 255.8 and 276.0 mm, and Yeongsan River was 172.3 and 167.0 mm, respectively. The discharge for June and July of the Nakdong River was $8.6 \times 10^3$ and $22.0 \times 10^3$ m$^3$ s$^{-1}$, Seomjin River was $1.5 \times 10^3$ and $6.6 \times 10^3$ m$^3$ s$^{-1}$, and Yeongsan River was $2.0 \times 10^3$ and $3.1 \times 10^3$ m$^3$ s$^{-1}$, respectively. The Nakdong River had higher precipitation and discharge values compared to the other rivers.

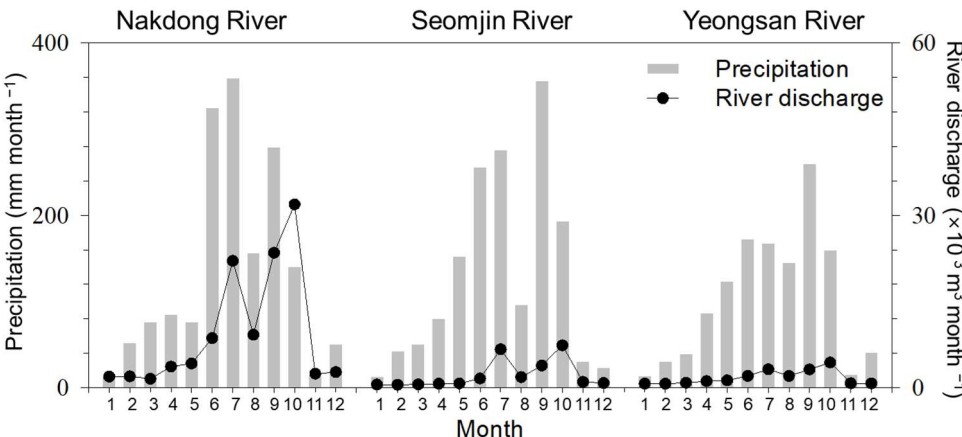

**Figure 2.** Monthly precipitation (bars) and discharge flow (points) at the three rivers (Nakdong, Seomjin and Yeongsan) in 2019. The gray bar represents precipitation, and the black dots connected with the black line represent river discharge.

### 3.2. Abiotic Factors in KCWs

The horizontal distribution of environmental factors between and within zones varied (Figure 3). In the SI zone, the water temperature in the surface layer ranged from 16.75 to 25.18 °C. Within the SI zone, the water temperature was the highest at St. SI 12 and 13 (near the Nakdong River), and the lowest at St. SI 3. In the SO and YS zones, the water temperatures ranged from 21.8–24.29 °C and 22.63–24.93 °C, respectively, and there was no significant difference between these zones.

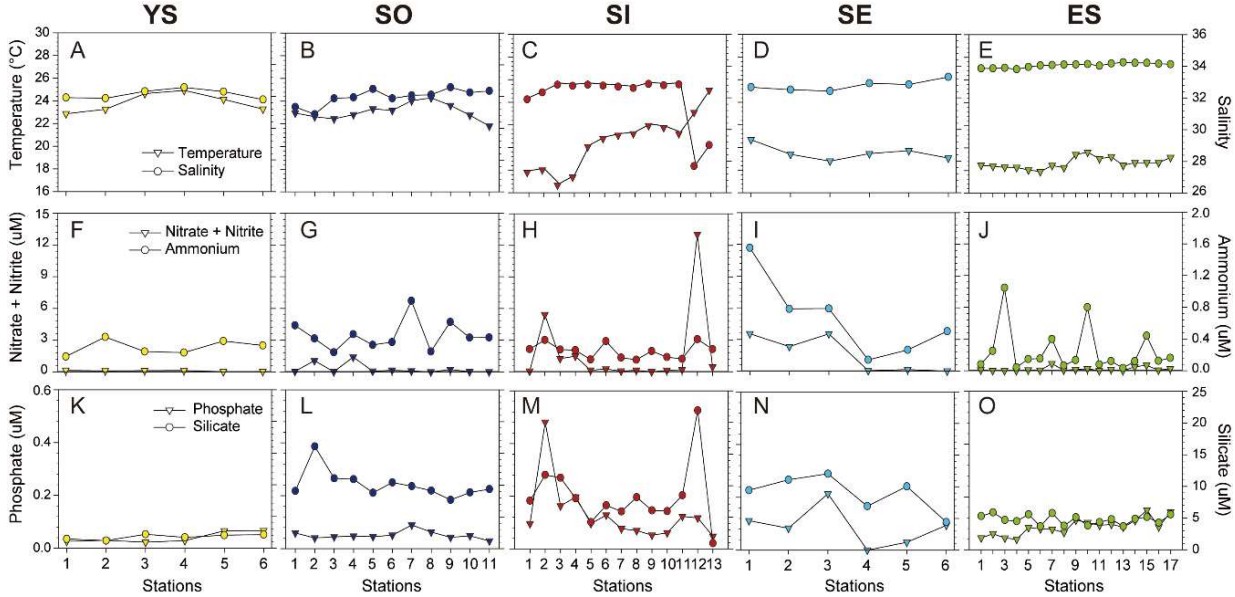

**Figure 3.** Horizontal distribution of environmental factors such as temperature and salinity measured by YSI sonde (**A**–**E**), nitrate + nitrite and ammonium (**F**–**J**), and phosphate and silicate (**K**–**O**) by zone in Korean Coastal Waters(KCWs). The triangle and cricle are color coded by zone (yellow: YS = Yellow Sea, dark blue: SO = Southern Offshore, red: SI = Southern Inshore, light blue: SE = Southeastern area, green: ES = East Sea).

Water temperatures in SE ranged from 18.80–20.70 °C, and in ES ranged from 17.91–19.62 °C. The average water temperatures of the SE and ES zones were not statistically different. According to satellite data in June, water temperatures in the ES zone and the southern Korean coast were relatively cold compared to offshore waters. The low water temperatures observed in the vicinity of Jindo and Cheongsando persisted until July (Figure 4).

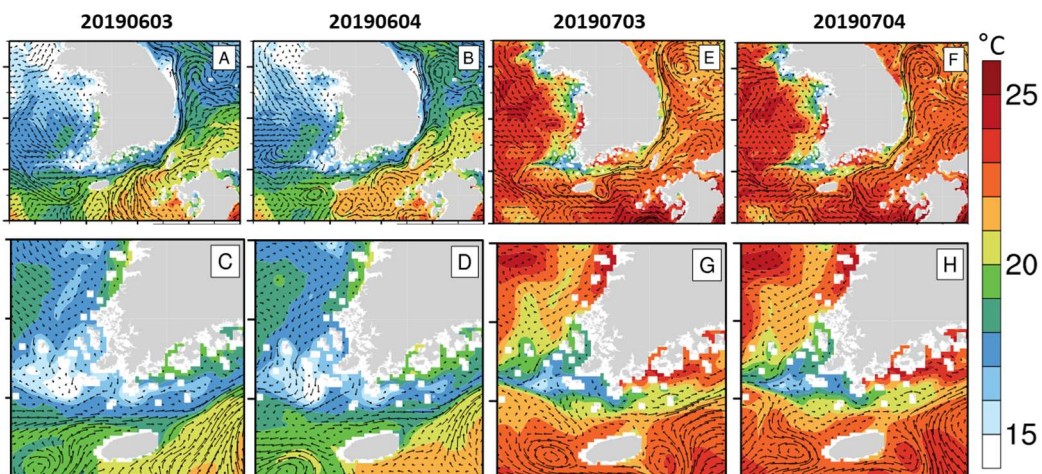

**Figure 4.** Horizontal distributions for the daily surface temperature (SST) with the surface currents obtained from the ocean analysis data (OPEM) from KIOST during 3 June 2019 (**A,C**), 4 June 2019 (**B,D**), 3 July 2019 (**E,G**), and 4 July 2019 (**F,H**). The Korean Peninsula (**A,B,E,F**), and the Yongsan River Estuary (**C,D,G,H**).

The surface salinity ranged from 27.75–32.95 at SI, 31.89–32.59 at SO, 30.89–32.57 at YS, 32.43–33.32 at SE, and 33.82–34.25 at ES. The surface salinities were consistently high (>31.88) in most areas. However, St. SI 1 and 2 had low salinities of less than 29 due to discharge from the Nakdong River. In June, a water mass of low salinity was observed in the western region of the East China Sea, and this water mass gradually extended based on the satellite surface salinity (Figure 5). As a result, the southwestern coast of Jeju Island became less saline in July.

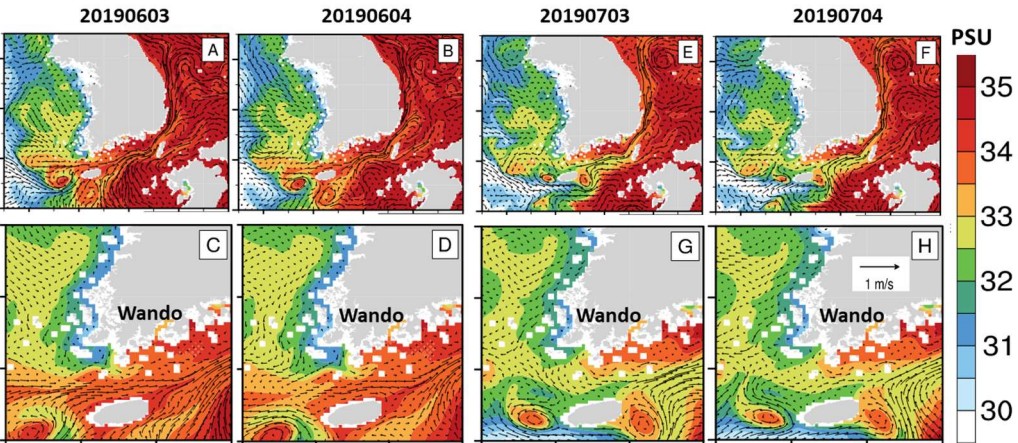

**Figure 5.** Horizontal distributions for the daily surface salinity (SSS) with the surface currents obtained from the ocean analysis data (OPEM) from KIOST, during 3 June 2019 (**A,C**), 4 June 2019 (**B,D**), 3 July 2019 (**E,G**), and 4 July 2019 (**F,H**). The Korean Peninsula (**A,B,E,F**), and the Yongsan River Estuary (**C,D,G,H**).

The nitrate + nitrite concentrations ranged from 0 to 13.05 µM at SI, 0–1.40 µM at SO, 0.02–1.06 µM at YS, 0–3.54 µM at SE, and 0–0.67 µM at ES. St. SI 2 and 12 had high concentrations of 13.05 µM and 5.43 µM, respectively. St. SE 1–3 (closest to the Nakdong River) had higher concentrations compared to the other stations in the same zone that averaged 3.13 ± 0.57 µM. The concentration of ammonium ranged from 0.15 to 0.41 µM at SI, 0.25–0.90 µM at SO, 0.19–0.59 µM at YS, 0.14–1.56 µM at SE, and 0.03–1.05 µM at ES. In the SE zone, all the stations were relatively low in ammonium (<1 µM), except for St. SE 1, which had the highest concentration of 1.56 µM. In ES zone, there were four peaks at stations 3, 7, 10 and 15 of 1.05, 0.40, 0.80 and 0.44 µM, respectively. The concentration

of phosphate ranged from 0.05 to 0.48 μM at SI, 0.03–0.09 μM at SO, 0.02–0.07 μM at YS, 0–0.21 μM at SE, and 0.04–0.15 μM at ES. The phosphate concentrations in the SI zone were extremely low with an average of 0.1 ± 0.04 μM, except for St. SI 12, which was the highest at 0.48 μM. The silicate concentration ranged from 0.88 to 21.91 μM at SI, 7.67–11.07 μM at SO, 1.22–16.10 μM at YS, 4.42–12.08 μM in SE, and 3.69–5.89 μM at ES. Similarly, the nitrate + nitrite concentrations and silicate levels were high at St. SI 2 (21.91 μM) and 12 (11.71 μM).

The ratios of nutrient concentrations suggest N limitation in most zones (Figure 6). The average N/P ratios were 6.6 ± 3.9 at SI, 15.8 ± 11.4 at SO, 12.3 ± 4.9 at YS, 25.2 ± 13.7 and 4.9 ± 5.7 at ES; N limitation was most pronounced at ES. The Si/P ratio was greater than 16 at all stations in KCWs, thus silicate was not a limiting nutrient.

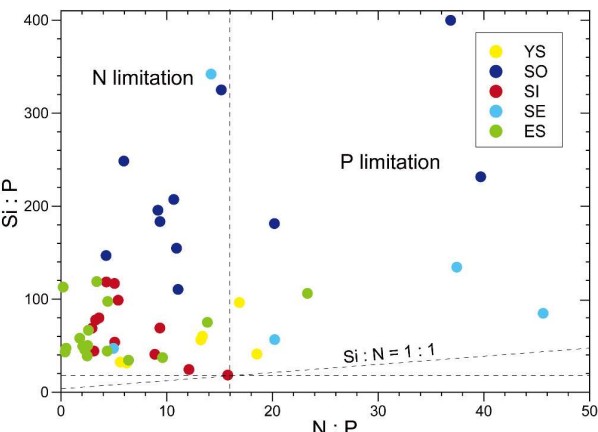

**Figure 6.** Si:N:P ratios of all stations in this survey. Points are color coded by zone and dashed lines indicate the optimal ratios for phytoplankton growth. Note that most stations were N limited.

The Chl *a* concentration ranged from 0.5 to 9.2 μg L$^{-1}$ in SI, 1.0–2.2 μg L$^{-1}$ in SO, 1.0–2.0 μg L$^{-1}$ in YS, 0.16–0.90 μg L$^{-1}$ in SE, and 0.01–0.18 μg L$^{-1}$ in ES (Figure 7). In SI, the spatial variation in Chl *a* was relatively large, with the highest values of 8.8 and 9.2 μg L$^{-1}$ at stations 11 and 12, respectively (Figure 8). In SE, phytoplankton abundance was high, but Chl *a* concentrations were significantly low, averaging 0.58 ± 0.28 μg L$^{-1}$ (ANOVA; $F = 8.587$, $p < 0.001$; Table 1). In ES, the Chl *a* concentrations were the lowest among all the zones, averaging 0.07 ± 0.05 μg L$^{-1}$.

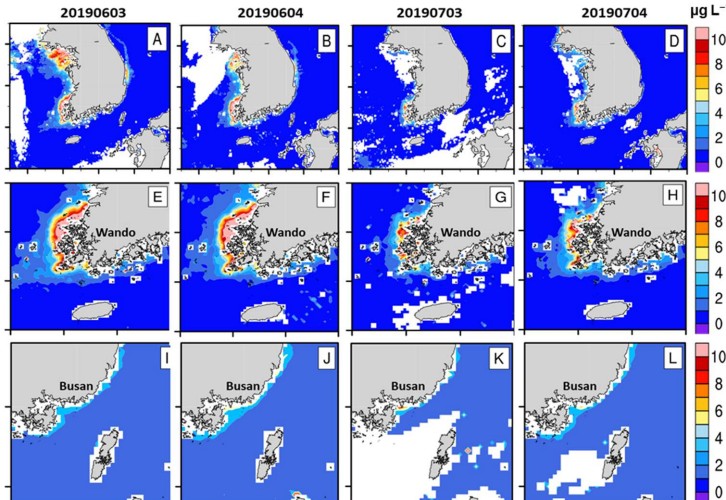

**Figure 7.** Horizontal distributions for Chl *a* obtained from the daily satellite observation data (JAXA Himawari–8 Level 3), from 3 June 2019 (**A,E,I**), 4 June 2019 (**B,F,J**), 3 July (**C,G,K**), and 4 July (**D,H,L**). The Korean Peninsula (top), the Yongsan River Estuary (middle), and the Nakdong River Estuary (bottom).

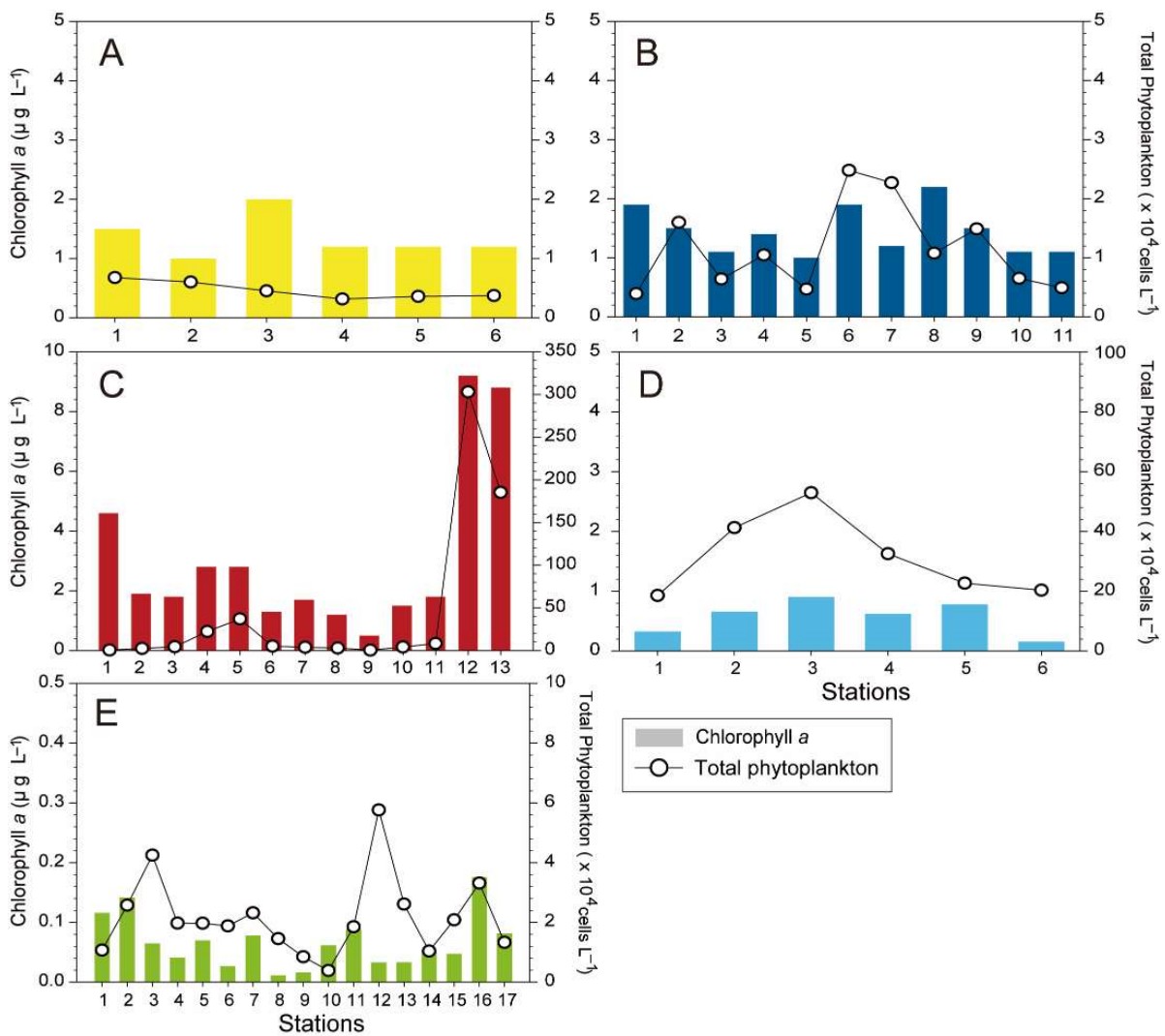

**Figure 8.** Horizontal distribution of chlorophyll *a* (bars) and total phytoplankton abundance (points) in YS (**A**), SO (**B**), SI (**C**), SE (**D**), and ES (**E**).

**Table 1.** Abiotic and biotic factors of all five zones during the summer of 2019 (YS = Yellow Sea; SO = Southern Offshore; SI = Southern Inshore; SE = Southeastern area; ES = East Sea). The data were averaged for each zone. Values are means ± standard errors, and results were compared using a one–way ANOVA with Tukey's honest significant difference (HSD) post-hoc test. Means with the same letter are not significantly different (*N.S*), * $p < 0.05$, ** $p < 0.01$, *** $p < 0.001$. Letters (a, b and c) indicate significantly differences among the five zones.

| | YS | SO | SI | SE | ES | *F*–Value |
|---|---|---|---|---|---|---|
| Temperature (°C) | 23.86 ± 0.84 [b] | 23.08 ± 0.72 [b] | 20.56 ± 2.43 [a] | 19.53 ± 0.66 [a,c] | 18.65 ± 0.48 [c] | 28.23 *** |
| Salinity | 32.13 ± 0.31 [a] | 31.99 ± 0.50 [a] | 32.07 ± 1.67 [a] | 32.78 ± 0.32 [a] | 34.07 ± 0.14 [b] | 14.31 *** |
| Chl *a* (µg L⁻¹) | 1.35 ± 0.36 [a,b] | 1.45 ± 0.40 [a,b] | 3.07 ± 2.81 [a] | 0.58 ± 0.28 [b] | 0.07 ± 0.05 [b] | 8.56 *** |
| Nitrate + Nitrite (µM) | 0.10 ± 0.05 | 0.28 ± 0.48 | 1.75 ± 3.70 | 1.59 ± 1.74 | 0.14 ± 0.20 | 1.86 [N.S] |
| Ammonium (µM) | 0.31 ± 0.10 [a] | 0.46 ± 0.019 [a,b] | 0.26 ± 0.09 [a] | 0.68 ± 0.51 [b] | 0.25 ± 0.28 [a] | 4.17 ** |
| Phosphate (µM) | 0.04 ± 0.02 [a] | 0.05 ± 0.02 [a,b] | 0.13 ± 0.11 [b] | 0.09 ± 0.07 [a,b] | 0.09 ± 0.03 [a,b] | 3.24 * |
| Silicate (µM) | 1.80 ± 0.40 [a] | 10.11 ± 2.23 [c] | 8.24 ± 4.96 [b,c] | 9.02 ± 2.85 [c] | 4.76 ± 0.75 [a,b] | 12.06 *** |
| Composition of phytoplankton (%) | | | | | | |
| Diatoms | 8.72 ± 16.05 [a] | 10.18 ± 8.28 [a] | 40.08 ± 34.58 [a,b] | 66.23 ± 9.00 [b] | 25.03 ± 24.46 [a] | 7.40 *** |
| Dinoflagellates | 60.99 ± 20.78 [c] | 45.74 ± 9.21 [b,c] | 21.60 ± 20.84 [a] | 10.00 ± 3.28 [a] | 23.51 ± 14.28 [a,b] | 14.38 *** |
| Cryptophytes | 18.37 ± 19.65 [a,b] | 4.17 ± 10.21 [a] | 27.16 ± 22.08 [b] | 16.98 ± 7.30 [a,b] | 16.08 ± 11.01 [a,b] | 2.23 [N.S] |
| Others | 11.92 ± 7.66 [a] | 39.91 ± 11.93 [b] | 11.16 ± 9.24 [a] | 6.79 ± 2.88 [a] | 35.38 ± 12.34 [b] | 22.48 *** |

### 3.3. Phytoplankton Community

We measured the total abundance and composition of phytoplankton in KCWs. In SI, the total phytoplankton abundance ranged from 0.5 to $303.0 \times 10^4$ cells $L^{-1}$ (average: $44.7 \times 10^4 \pm 92.3 \times 10^4$ cells $L^{-1}$). Diatoms (40%) were dominant at St. SI 2–6 and 13, and cryptophytes were dominant at St. SI 12, near the Nakdong River. The dominant diatom species were *Chaetoceros* spp., and *Cryptomonas* spp. at most stations, except for St. SI 5, where the dominant species was *Skeletonema* spp. In SO, the total phytoplankton abundance ranged from $0.4 \times 10^4$ to $2.4 \times 10^4$ cells $L^{-1}$ (average: $1.2 \times 10^4 \pm 0.7 \times 10^4$ cells $L^{-1}$). Dinoflagellates were dominant at all stations, and the dominant species were *Gyrodinium* spp. and *Gymnodinium* spp. at most stations, except for St. SO 1 and 2, where cryptophytes were most dominant. In YS, the total phytoplankton abundance ranged from $0.3 \times 10^4$ to $1.5 \times 10^4$ cells $L^{-1}$ (average: $0.5 \times 10^4 \pm 0.2 \times 10^4$ cells $L^{-1}$), and the abundances of phytoplankton were relatively low at most stations (Figure 9). Dinoflagellates were the most dominant and had the highest composition (ANOVA; $F = 14.382$, $p < 0.001$) compared to the other zones. The dominant species were *Katodinium* spp. in St. YS 1 and 2 and *Nitzschia* spp. in St. YS 3 to 5. In SE, the total phytoplankton abundance ranged from 18.5 to $52.9 \times 10^4$ cells $L^{-1}$ (average: $31.4 \times 10^4 \pm 13.6 \times 10^4$ cells $L^{-1}$). Diatoms were dominant, and *Chaetoceros* spp. and *Leptocylindrus danicus* were dominant at all stations. In ES, the total phytoplankton abundance ranged from 0.3 to $5.7 \times 10^4$ cells $L^{-1}$ (average: $2.2 \times 10^4 \pm 1.3 \times 10^4$ cells $L^{-1}$). Diatoms were dominant in St. ES 1, 3, 11 and 12, and dinoflagellates were dominant in St. ES 6, 13 and 15. *Cryptomonas* spp. dominated at most stations, except St. ES 11 and 12, where *Chaetoceros* spp. were dominant, and *Prorocentrum* spp. were present in high abundance in St. ES 13, 14, 15 and 17.

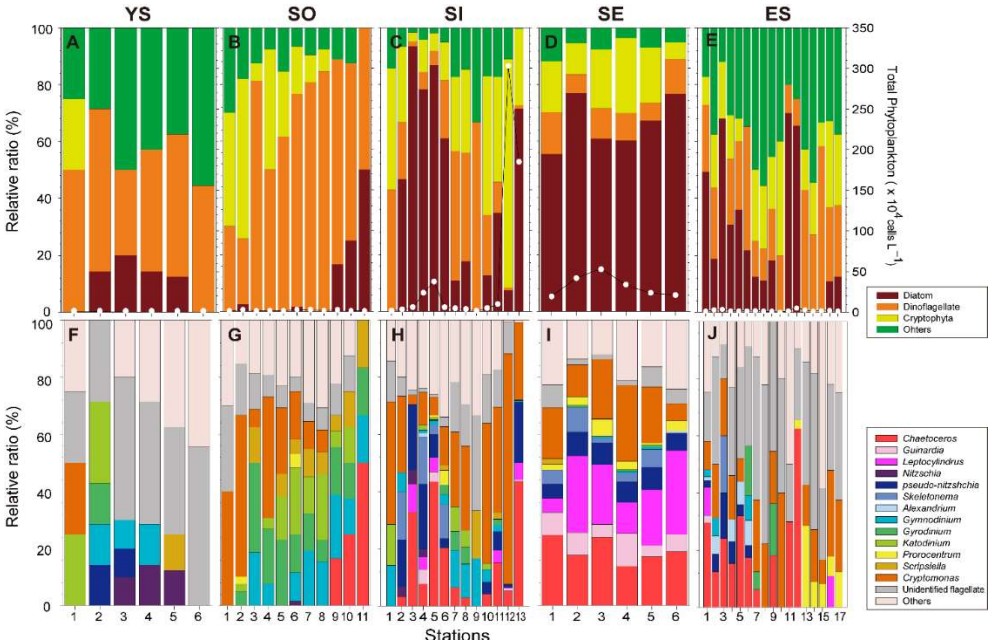

**Figure 9.** Total phytoplankton abundance and relative ratio of each group (**A**–**E**), and relative ratios of the dominant species across all the zones (**F**–**J**).

### 3.4. Statistical Analysis

We used PCA analysis to explore the relationship between abiotic and biotic factors in KCWs (Table 2). In SI, PC1 (which accounted for 42.10% of the variance) had positive loading factors for DIN, DSi, Chl *a* and cryptophytes, and a negative loading for salinity. PC2 (which accounted for 25.84% of the variance) had positive correlations with DIN (dissolved inorganic nitrogen) and diatoms, and negative correlations with Chl *a* and cryptophytes. In SO, PC1 (which accounted for 33.83% of the variance) had positive loading factors for DIN, DSi (dissolved silicate) and cryptophytes, and strong negative

loadings for salinity. PC2 (which accounted for 25.97% of the variance) had a positive loading for temperature, DIP (dissolved inorganic phosphate) and dinoflagellates, while there was no significantly negative loading. In YS, PC1 accounted for 41.06% of the variance, and temperature, salinity and DSi were positively correlated, while dinoflagellates and cryptophytes were negatively correlated with PC1. PC2 (which accounted for 25.84% of the variance) had positive loading factors for DIN and diatoms, and negative loading factors for Chl *a* and cryptophytes. In SE, PC1 (which accounted for 34.04% of the variance) had positive loadings for Chl *a*, diatoms and cryptophytes, and a negative loading for temperature. PC2 (which accounted for 34.00% of the variance) had positive correlations with DIN, DSi and Chl *a*, and a negative correlation with salinity. In ES, PC1 (which accounted for 25.55% of the variance) had positive loadings for temperature, salinity and DIP, and no negative loadings. PC2 (which accounted for 18.21% of the variance) had a positive correlation with dinoflagellates, and a negative correlation with diatoms.

**Table 2.** Loading factors of horizontal environmental variables and phytoplankton biomass by zone in KCWs for PC1 and PC2 (factors > 0.5 are in bold).

| Parameters | YS | | SO | | SI | | SE | | ES | |
|---|---|---|---|---|---|---|---|---|---|---|
| | PC1 | PC2 | PC1 | PC2 | PC1 | PC2 | PC1 | PC2 | PC1 | PC2 |
| Temperature (°C) | **0.91** | 0.12 | −0.22 | **0.83** | 0.07 | **0.84** | **−0.90** | 0.41 | **0.78** | −0.25 |
| Salinity | **0.76** | 0.02 | **−0.73** | 0.23 | **−0.72** | **−0.66** | −0.23 | **−0.89** | **0.77** | 0.37 |
| DIN (μM) | −0.25 | **0.96** | **0.83** | 0.18 | **0.99** | 0.02 | −0.28 | **0.56** | 0.49 | −0.37 |
| DSi (μM) | **0.77** | −0.44 | **0.89** | −0.25 | **0.91** | −0.31 | 0.28 | **0.87** | 0.07 | −0.07 |
| DIP (μM) | 0.13 | −0.13 | 0.05 | **0.82** | 0.36 | **−0.61** | 0.15 | 0.11 | **0.88** | 0.38 |
| Chl *a* (μg L⁻¹) | 0.32 | **−0.52** | −0.03 | 0.24 | **0.61** | **0.69** | **0.70** | **0.67** | −0.20 | 0.23 |
| Diatoms | 0.35 | **0.71** | −0.31 | −0.26 | 0.01 | **0.85** | **0.80** | 0.28 | −0.11 | **−0.65** |
| Dinoflagellates | **−0.95** | 0.21 | 0.01 | **0.85** | 0.42 | **0.75** | 0.47 | 0.19 | 0.01 | **0.83** |
| Cryptophytes | **−0.70** | **−0.60** | **0.94** | 0.01 | **0.89** | 0.41 | **0.65** | 0.47 | −0.18 | 0.02 |
| Eigenvalue | 3.70 | 2.33 | 3.04 | 2.34 | 3.79 | 3.53 | 3.74 | 3.74 | 2.30 | 1.64 |
| Variability (%) | 41.06 | 25.84 | 33.83 | 25.97 | 42.10 | 39.17 | 34.04 | 34.00 | 25.55 | 18.21 |
| Cumulative (%) | 41.06 | 66.90 | 33.83 | 59.80 | 42.10 | 81.27 | 34.04 | 68.04 | 25.55 | 43.76 |

The CCA results indicated that variables such as temperature, ammonium, nitrate + nitrite, salinity, Chl *a*, silicate and phosphate can explain the variability of phytoplankton species (Figure 10). In particular, *Skeletonema* spp. had the strongest relationship with phosphate, and *Cryptomonas* spp. had a negative correlation with salinity and had positive correlations with most of the nutrients (nitrate + nitrite, ammonium, silicate) in SI. In SO, *Cryptomonas* spp. also had negative correlation with salinity and positive correlations with nitrate + nitrite and silicate.

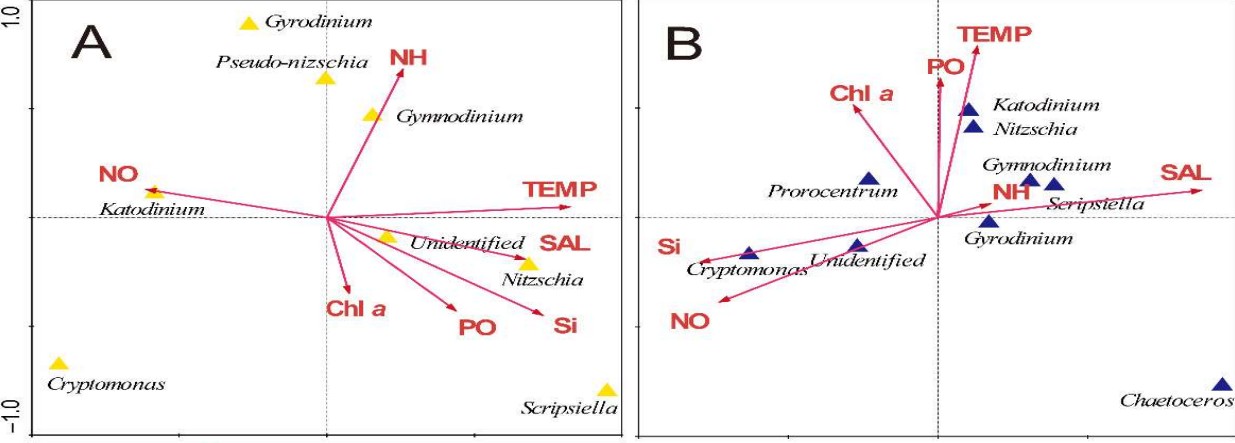

**Figure 10.** *Cont.*

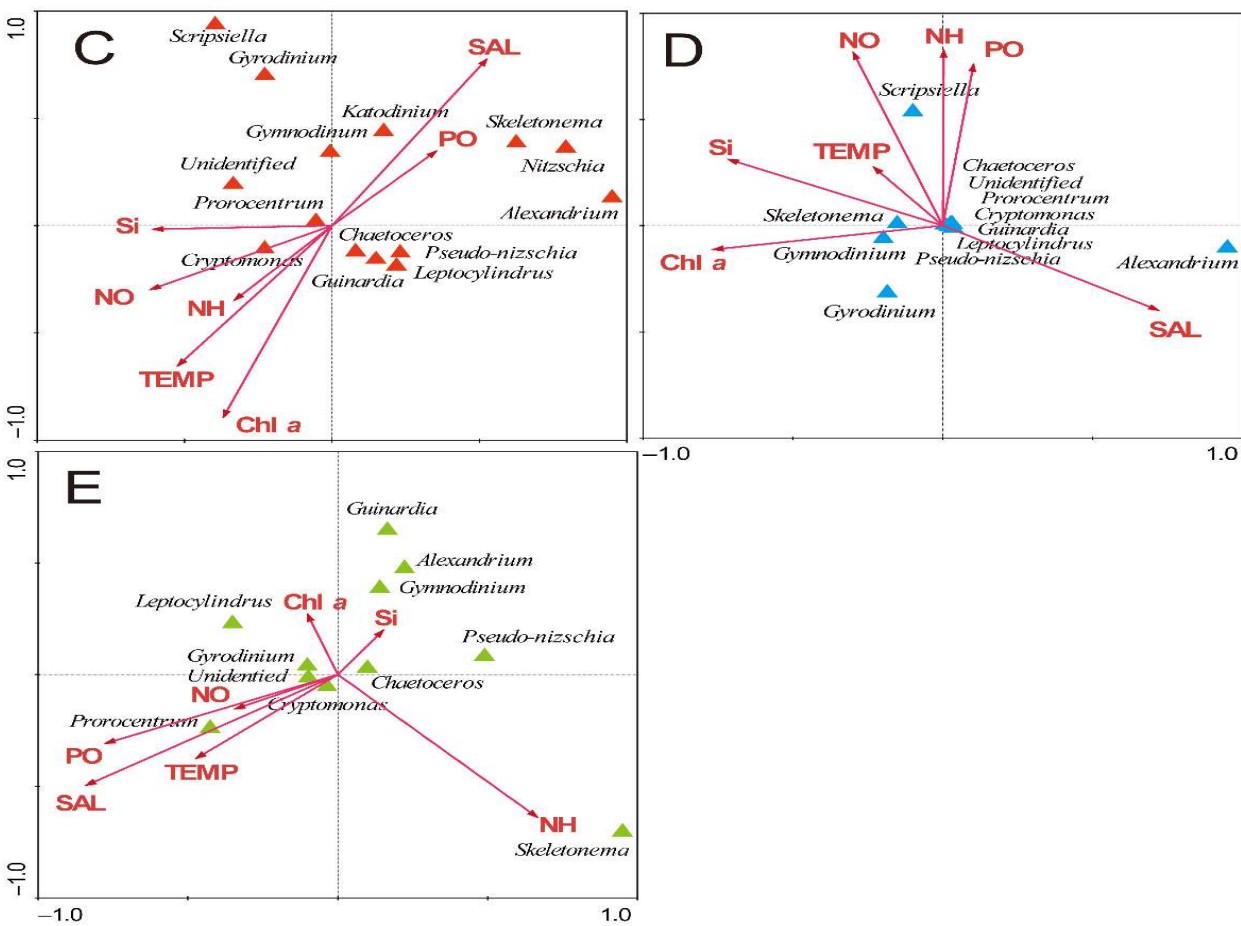

**Figure 10.** Canonical correspondence analysis (CCA) of the relationships between the dominant phytoplankton (triangle) and environmental factors (arrows; TEMP: water temperature, SAL: salinity, Chl *a*: chlorophyll *a*, NO: nitrate + nitrite, NH: ammonium, Si: silicate, and PO: phosphate) of YS (**A**), SO (**B**), SI (**C**), SE (**D**), and ES (**E**).

## 4. Discussion

### 4.1. Environmental Characteristics in the KCWs

It is well known that large amounts of Changjiang River discharge strongly affect the East China Sea [16]. This low salinity (<30) water masses is known as Changjiang Diluted Water (CDW), and significantly dominates during the summer monsoon season in East Asia. The impact of CDW gradually extends from the East China Sea to the East Sea (Sea of Japan) [17]. In the present study, we observed a relatively low salinity at St. SO 1 and 2, and these stations are in the CDW inflow path. Based on satellite data, we confirmed that the low–salinity water mass gradually expanded into southwestern KCWs including Jeju Island from June to July. It is unlikely that this low salinity observed in the offshore waters of St. SO 1–2 is a result of freshwater input from the Yeongsan River located in the southwestern region of Korea, since the discharge volume is low and there is a strong intrusion of saline water from the Jeju Warm Current [18]. Instead, the low–salinity water masses can likely be attributed to the effect of CDW. According to Suh et al. [19], the creation of a low–salinity environment by massive CDW intrusion causes mass mortality of valuable macro–benthic organisms such as abalone and conch near Jeju Island. On the other hand, Lim et al. [20] suggested the possibility that the low salinity and high temperature of CDW may have negative effects on harmful algal bloom (HAB) species such as *Cochlodinium polykrikoides*; this species is responsible for massive fish mortality in Korea during the summer [21]. The CDW has low salinity, high temperature, and a high concentration of nutrients, especially DIN [20,22,23]. Nutrient and salinity changes related to the intrusion of the CDW during



the summer play an important role in determining the phytoplankton community and population dynamics, such as inhibiting some HABs.

In the southern KCWs, the relatively narrow and shallow Korea Strait connects the East China Sea and the East Sea, and is influenced by the Tsushima Warm Current (TWC), which is derived from the Kuroshio Current. In particular, southwestern KCWs are significantly influenced by the Yellow Sea Current and the Jeju Warm Current [24]. In temperate regions, stratification of the water column develops gradually during spring and intensifies during summer. Under a stratified water column, nutrients are not well supplied to the euphotic zone, maintaining the low nutrient concentration in the surface layers during summer. In the present study, high nutrient concentrations were maintained between St. SI 2–4 and SI 12. The supply of nutrients in surface layers of western KCWs was influenced by vertical water mixing, such as upwelling events [25–27]. Wind–driven coastal upwelling transfers low–temperature and nutrient–rich waters from the bottom into the euphotic zone after 2–3 days of strong southerly winds in south–eastern KCWs (i.e., St. SE 5 and 6), and strong northerly winds in south–western KCWs (i.e., St. SI 2–4) [1,18,28]. During this study period, the northern winds were dominant, and the surface water temperature was significantly lower near St. SI 2–4 (*t*-test: $t = -5.31$, $p < 0.05$) compared to other stations in field surveys and satellite observations. In particular, the water temperatures near the coast of Jindo were low, and the concentrations of silicate and phosphate were high. Therefore, our field survey and satellite–based horizontal profiles can support the presence of an upwelling.

In southern KCWs, nutrient loading from the Nakdong Rivers usually peaks in summer, probably due to a large amount of river discharge after heavy rains in rainy seasons [1]. During the study period, the observed salinity was significantly low (*t*-test: $t = 6.434$, $p < 0.05$) in St. SI 12, and thus high nutrients were recorded around the location in July; on the other hand, low salinity water masses were not observed in the SE zone in June. Although there was no significant difference in the amount of precipitation between June and July 2019, the river discharge in July was about two times higher than that of June. Since there are several artificially constructed barrages in the Nakdong River [29], we speculate that the salinity difference is due to the adjustment of the discharge amount depending on whether the dams are open or not. The Nakdong River plays an important role in supplying a large amount of nutrients to the southeastern KCWs, particularly DIN and DSi [29,30]. Therefore, we suggest that the control of freshwater inflow according to the operation of barrages can have profound effects on the community structure of phytoplankton, which responds sensitively to salinity and nutrients, as mentioned below in detail.

*4.2. Phytoplankton Community and Distribution in the KCWs*

It is well known that nutrient loading from river discharge increases the abundance of phytoplankton [31,32]. Changes in salinity and nutrients related to upwelling events and freshwater inflow were found in this study, which can lead to changes in the phytoplankton community. In the upwelling area (St. SI 2–4) of western KCWs, diatoms have significantly high ratios ($73 \pm 19\%$; average: $7.6 \times 10^4 \pm 8.7 \times 10^4$ cells $L^{-1}$) compared to other nearby locations ($22 \pm 26\%$; average: $1.0 \times 10^4 \pm 1.4 \times 10^4$ cells $L^{-1}$). Among them, *Skeletonema* spp. was particularly dominant. On the other hand, eastern KCWs were influenced by freshwater discharge. In low–salinity and high–nutrient conditions, there was a significantly higher ratio of *Cryptomonas* spp. (81%) compared to diatoms (8%) at St. SI 12. Phosphate availability limits the growth of diatoms more so than flagellates [33], and even in a nitrate–rich environment, the growth of diatoms is limited since they do not use nitrates [34]. The upwelling areas (St. SI 2–4) had a significantly higher phosphate (*t*-test: $t = -3.807$, $p < 0.05$) concentration than areas with freshwater input (St. SI 12). It is considered that the nutrient composition of the upwelling area had a positive effect on the growth of the *Skeletonema* spp. of diatom, which is also explained by the strong positive correlation between cell density and phosphate concentration of *Skeletonema* spp. based on the CCA results (Figure 10).

*Cryptomonas* spp. has a wide tolerance to changes in the environment (water temperature and salinity) [35,36], and is known to grow rapidly and dominate in water masses where there is no competition for nutrients [37,38]. Therefore, *Cryptomonas* spp. are dominant in regions where salinity is low and nutrient concentrations are high [39], such as upstream in the Seomjin River [40], and the southern coast of Korea. *Cryptomonas* spp. were not only found in regions with high freshwater impact (St. SI 12), but also in less saline regions affected by the CDW (St. SO 1 and 2). According to our PCA and CCA results, *Cryptomonas* spp. and salinity had a strong negative correlation (Figure 10 and Table 1). Meanwhile, St. SI 12 and SO 1–2 mostly consisted of diatoms, and of the diatom species, *Chaetoceros* spp. was most dominant. It has been reported that *Chaetoceros* spp. is dominant in regions with higher salinity like the Busan coastal regions to avoid low salinity stress [30]. Moreover, the specific growth rate of *Chaetoceros convoltus* and *Chaetoceros concavicorinis* decreased when the salinity was lower than 30 [41]. Therefore, these findings suggest that the relatively high adaptability of *Cryptomonas* spp. to a low–salinity environment may confer a competitive advantage over *Chaetoceros* spp. in low–salinity waters affected by the Nakdong River.

The diatom *Leptocylindrus danicus* was dominant in the region affected by the Nakdong River (SE zone). *L. danicus* is a major species that frequently occur in coastal waters [42], and growth is not inhibited in environments with salinities below 31 [43]. The nutrient concentrations in SE during June were high due to freshwater inflow, but maintained a salinity of 31 or higher with a lower river discharge compared to July. In addition, the surface water temperature during the study period was 15–20 °C, which is known as the optimum growth water temperature for *L. danicus* [44]. Therefore, the combination of high nutrients and salinity, and the optimal water temperature range had a profound effect in allowing *L. danicus* to dominate.

Picoplankton, despite its small cell size, plays crucial roles in marine ecosystems related to primary production [45,46]. In ES and YS zones, unidentified small flagellates (35, 40%) appeared at a high rate. Recently, it has been reported that cyanobacteria are present in high proportions during the summer off the coast of Dokdo and in the Yellow Sea [47]. The concentrations of nitrate play an important role in phytoplankton growth [48], and according to Dorth and Whitledge [49], phytoplankton growth was significantly lower when the concentrations of DIN and DIP were below 1.0 and 0.2 μM, respectively. In the ES (DIN: $0.39 \pm 0.40$ μM, DIP: $0.09 \pm 0.03$ μM) and YS (DIN: $0.40 \pm 0.07$ μM, DIP: $0.04 \pm 0.02$ μM) zones, the concentration of DIN and DIP was lower than this standard. Small cell microalgae are favored in oligotrophic water masses due to their high nutrient uptake relative to cell surface area [50,51]. In the low nutrient conditions of the East Sea (DIN: <0.5 μM, DIP: <0.1 μM), picoplanktonic cyanobacteria such as *Prochlorococcus* and *Synechococcus* were dominant [52–54]. Similarly, in the Yellow Sea, the major phytoplankton component was cyanobacteria in nutrient–depleted conditions during the summer [55]. Recently, cyanobacteria including *Prochlorococcus* and *Synechococcus* were dominant under DIN–limited conditions in the surface waters of the East Sea [47], which are favorable for the growth of these taxa [56,57]. In this study, the average N/P ratio was <13 in the ES and YS zones (N-limitation; Figure 6) and had low nutrient levels, indicating that during summer, the environments of the ES and YS zones are favorable for the growth of these organisms. Therefore, there is a possibility that the high proportions of unidentified small flagellates in this study are *Prochlorococcus* and *Synechococcus*; however, this needs to be confirmed by advanced analysis methods such as genetic sequencing or pigment analysis. In addition, small sized phytoplankton (<5 μm) have a negative effect on the carbon energy flow of marine pelagic organisms [58]. As a result, we can consider that small–sized phytoplankton related to low Chl *a* in the ES and YS zones may lead to the reduced transfer of energy toward higher trophic levels, particularly in the euphotic zone during summer [59].

## 5. Conclusions

We analyzed environmental variables in KCWs to better understand the ecological dynamics of phytoplankton related to CDW spreading, upwelling, and freshwater runoff from the Nakdong River. In particular, the changes in the phytoplankton community structure were influenced by hydro–oceanographic events that provide nutrient rich waters into the coastal zone by upwelling (St. SI 2–4), CDW in southwestern KCW, and freshwater runoff (St. SI 12) from the Nakdong River in southeastern KCWs. As a result, cryptophytes and diatoms were maintained in high nutrients during the summer in KCWs. On the other hand, the East Sea (DIN: $0.39 \pm 0.40$ μM, DIP: $0.09 \pm 0.03$ μM) and Yellow Sea (DIN: $0.40 \pm 0.07$ μM, DIP: $0.04 \pm 0.02$ μM) were characterized by low nutrient levels and low Chl *a,* which can lead to a dominance of unidentified small flagellates rather than diatoms. Therefore, our field survey and satellite–based horizontal profiles can contribute to the explanation of phytoplankton population dynamics related to hydro–oceanographic events during the summer in KCWs.

**Author Contributions:** Conceptualization, J.N.Y. and S.H.B.; Data curation, M.L., H.J. and Y.G.P.; Formal analysis, M.L.; Funding acquisition, S.H.B.; Visualization, J.N.Y., M.L., H.J. and Y.G.P.; Writing—original draft, J.N.Y.; Writing—review & editing, J.N.Y., Y.K.L., H.R. and S.H.B. All authors have read and agreed to the published version of the manuscript.

**Funding:** This work was supported by the project "a sustainable research and development of Dokdo (PG52911)" of the Ministry of Oceans and Fisheries. This research was also supported by the Ministry of Oceans and Fisheries of Korea [20220357; Land/Sea–based input and fate of microplastics in the marine environment] and the KIOST project (PEA0016).

**Institutional Review Board Statement:** Not applicable.

**Informed Consent Statement:** Not applicable.

**Data Availability Statement:** The data presented in this study are available on request from the corresponding author.

**Acknowledgments:** We would like to acknowledge the captains and crew onboard the R/V Eardo and R/V Onnuri for making sampling possible. The research product of chlorophyll *a* (produced from Himawari-8) that was used in this paper was supplied by the P-Tree System, Japan Aerospace Exploration Agency (JAXA).

**Conflicts of Interest:** The authors declare no conflict of interest.

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
