# Peer review of "Summer Distributional Characteristics of Surface Phytoplankton Related with Multiple Environmental Variables in the Korean Coastal Waters"

_jmse, doi:10.3390/jmse10070850_

Round 1
Reviewer 1 Report
Review for the paper "Summer distributional characteristics of phytoplankton related with multiple environmental variables in the Korean coastal waters" by Ji Nam Yoon, Minji Lee, Hyunkeun Jin, Young Kyun Lim, Hyejoo Ro, Young-Gyu Park, Seung Ho Baek submitted to "Journal of Marine Science and Engineering".
General comment.
In the marine environment, plankton populations exhibit considerable variability on a broad range of timescales. Marine phytoplankton are a diverse group of pelagic photosynthetic microbes that provide over 90% of marine primary production. Although marine phytoplankton account for only 0.2% of global photosynthetic carbon biomass, they generate >45% of the primary production. One of the most fundamental questions in phytoplankton research is to understand environmental impacts on the assemblages of microalgae. In this article, the authors attempt to present the spatio-temporal variability of phytoplankton during the summer period of 2019 in the Korean coastal waters. Also, the present study examined the responses of phytoplankton to changes in a set of environmental parameters. The topic sounds interesting and could have contributed to better understanding the significance of different physical, chemical and biological parameters as sources of variability for the phytoplankton in the Korean coastal waters. Standard methods to collect samples and to threat the data were used in the study. Main results are illustrated with relevant Figures and Tables. Discussion is comprehensive and focused on the main findings. Statistical methods are adequate and correctly used. After minor revision this paper may be accepted for publication in "Journal of Marine Science and Engineering".
Specific remarks.
Title. The authors were focused on the surface phytoplankton. Therefore, I suggest slightly modifying the title as follows: Summer distributional characteristics of surface phytoplankton…
Abstract.
I recommend excluding detailed statistical description from this section (e.g. (t-test: t = 6.434 etc). You may use general statement such as p<0.05.
Additional information regarding phytoplankton needs to be present in the Abstract (contribution of dominant taxa, abundance, and biomass).
L20. Missed full stop after ‘St. SI 12’.
L23. Consider replacing "Cryptomonas spp.." with " Cryptomonas spp.".
L27. Abbreviations for nutrients should be explained.
Introduction.
L52. Consider replacing "and an important "with "and is an important".
L53. Consider replacing "There is strong" with "There is a strong".
Methods.
L166-168. The authors must provide more detailed description regarding CCA. Indicate why CCA was chosen. Also, describe response and predictor variables. What about normalization/standardization of the data?
Results
L230-245. Fig. 9 not cited. Please, correct and also check and correct order of Figures in the ms. The current Fig. 9 must be Fig. 6.
L251. Consider replacing "Table2" with "Table 1".
Also, this Table 1 should be placed earlier in the ms at P8.
L289-312. Table 1 must be Table 2.
Discussion.
L356. Consider replacing "of below 30" with "locating below 30 m".
L364. Consider replacing "and a strong" with "and there is a strong".
L381. Consider replacing "striated" with "stratified".
L386. Consider replacing "introduces" with "transfers".
L411-412. Provide relevant references.
L417-418. Consider replacing "received the influence of freshwater" with "was influenced with the freshwater discharge".
L418. Consider replacing "In low-salt and high-nutrient conditions had a significantly" with "In low-salt and high-nutrient conditions, there was a significantly".
L429. Consider replacing "and known" with "and is known".
L437. Consider replacing "is dominant regions" with "is dominant in regions"
Author Response
Thank you for your attentive review. We did our best to revise the revisions of the manuscript. The answer to your comment is on page 19- 21.
Please see the attachment.

Reviewer 2 Report
Manuscript ID jmse-1773290
Article: Summer distributional characteristics of phytoplankton related with multiple environmental variables in the Korean coastal waters
Dear authors,
Thank you for allowing me to review your work on how phytoplankton communities relate to ocean currents, freshwater runoff, and upwelling and differ among zones. These are important factors that may drive the production of phytoplankton, and may vary by zones.
The manuscript is very well written. I like the way the authors explain the problem and why the study is important in the first paragraph of the Introduction.
1. One suggestion, but not a requirement, would be to add another map in the results section that has text on the map to describe the general conditions in each region as determined from your study. For example, YS and ES [low nutrient, low chlorophyll a, small flagellates].
2. As a fish biologist, I would also like to hear more about the importance of these findings on higher trophic levels, perhaps in the Discussion. For example, who eats small flagellates in the YS and ES?
3. I am also curious whether 2019 was an anomalously warm or cool year in the region, and how results may changes under future warming that might impact ocean currents, runoff, and upwelling. Maybe elaborate in the Discussion.
Overall, I enjoyed reading about conditions on the other side of the Pacific Ocean. Below are a few minor comments.
Line 19: Please define acronyms on first use. St. SO, SI.
Line 20. Is there a period missing after SI 12?
Line 21. Please modify this sentence. CDW have gradually expanded from ________ to southwestern KCWs from June to July
Line 23. Please modify this sentence to “dominance of Cryptomonas spp., a freshwater and brackish water algae”.
Line 24. Please modify the sentence to “dominated by oceanic diatoms Skeletonema costatum-“
Line 27-28. Please define DIN, DIP, ES, YS
Line 30. Did ocean currents influence the phytoplankton community structure? If not, then reword to
“such as upwelling and freshwater run-off, but not ocean currents”.
Line 36. I suggest adding freshwater “key component of freshwater, estuarine,”
Line 81. Delete harmful algal bloom (HABs). HAB has already been defined on line 41.
Line 125. I recommend adding a reference for Korea Institute of Ocean Science and Technology (KIOST).
Line 133. Add a space after 5.
Line 145. The link (http://www.argodatamgt.org/Access-to-data/Access-via-FTP-on-GDAC) results in an error ‘Module not Found’. Is there a more direct link to the data?
Line 168. Is there a reference for CANOCO version 4.5 for Windows? If so then please add.
Line 187. I am surprised to see how warm the waters are around Korea and Japan! Was 2019 an anomalously warm year relative to recent years? Was it a warmer than average year?
Line 295. Pleas define DIN on first use on line 292 rather than 295.
Line 371. Is there a reference for the massive fish mortality in Korea during summer?
Line 371. Change text to “massive fish mortality in Korea during summer”. It is common to put the time component of a sentence at the very beginning or end of a sentence.
Line 404. I am not familiar with “barrages”. Would it be useful to define here?
Line 419. I would delete one of the “81%” in the text “higher ratio of 81% of Cryptomonas spp. (81%)” .
Figure 1. Add Dokdo and Ulleungdo to Figure 1 for reverence to line 93.
Figure 1. Add EKWC to the map.
Figure 1. Please define: EKCC, WKCC, CCC, and TSWC in the caption.
Figure 1. Please add Changjiang River to the map for the readers’ reference on line 55.
Author Response
Thank you for your attentive review. We did our best to revise the revisions of the manuscript. The answer to your comment is on page 19 -23.
Please see the attachment.
